# Steroid receptor coactivator-1 modulates the function of Pomc neurons and energy homeostasis

Yongjie Yang[1], Agatha A. van der Klaauw[2], Liangru Zhu[1,3], Tessa M. Cacciottolo [2], Yanlin He [1], Lukas K.J. Stadler[2], Chunmei Wang[1], Pingwen Xu[1], Kenji Saito[1], Antentor Hinton Jr.[1], Xiaofeng Yan[1], Julia M. Keogh[2], Elana Henning[2], Matthew C. Banton[2], Audrey E. Hendricks[4,5], Elena G. Bochukova[2], Vanisha Mistry[2], Katherine L. Lawler [2], Lan Liao[6], Jianming Xu[6], Stephen O'Rahilly [2], Qingchun Tong [7], UK10K Consortium, Inês Barroso [4], Bert W. O'Malley [6], I. Sadaf Farooqi [2] & Yong Xu [1,6]

Hypothalamic neurons expressing the anorectic peptide Pro-opiomelanocortin (Pomc) regulate food intake and body weight. Here, we show that Steroid Receptor Coactivator-1 (SRC-1) interacts with a target of leptin receptor activation, phosphorylated STAT3, to potentiate Pomc transcription. Deletion of *SRC-1* in Pomc neurons in mice attenuates their depolarization by leptin, decreases *Pomc* expression and increases food intake leading to high-fat diet-induced obesity. In humans, fifteen rare heterozygous variants in *SRC-1* found in severely obese individuals impair leptin-mediated Pomc reporter activity in cells, whilst four variants found in non-obese controls do not. In a knock-in mouse model of a loss of function human variant (SRC-1[L1376P]), leptin-induced depolarization of Pomc neurons and *Pomc* expression are significantly reduced, and food intake and body weight are increased. In summary, we demonstrate that SRC-1 modulates the function of hypothalamic Pomc neurons, and suggest that targeting SRC-1 may represent a useful therapeutic strategy for weight loss.

[1] Children's Nutrition Research Center, Department of Pediatrics, Baylor College of Medicine, One Baylor Plaza, Houston, TX 77030, USA. [2] University of Cambridge Metabolic Research Laboratories, and NIHR Cambridge Biomedical Research Centre, Wellcome Trust-MRC Institute of Metabolic Science, Addenbrooke's Hospital, Cambridge CB2 0QQ, UK. [3] Division of Gastroenterology, Union Hospital, Tongji Medical College, Huazhong University of Sciences & Technology, Wuhan 430022, China. [4] Wellcome Sanger Institute, Cambridge CB10 1SA, UK. [5] Mathematical and Statistical Sciences Department, University of Colorado – Denver, Denver, CO 80204, USA. [6] Department of Molecular and Cellular Biology, Baylor College of Medicine, Houston, TX 77030, USA. [7] Brown Foundation Institute of Molecular Medicine, University of Texas Health Science Center at Houston, Houston, TX 77030, USA. These authors contributed equally: Yongjie Yang, Agatha A. van der Klaauw, Liangru Zhu, Tessa M. Cacciottolo. A full list of consortium members appears at the end of the paper. Correspondence and requests for materials should be addressed to I.S.F. (email: isf20@cam.ac.uk) or to Y.X. (email: yongx@bcm.edu)

Transcriptional coactivators and corepressors regulate the ability of nuclear hormone receptors (NRs) and transcription factors (TFs) to enhance/suppress the expression of target genes by facilitating the assembly of the transcription complex at target gene promoters[1]. Understanding the molecular mechanisms by which coactivators and corepressors alter gene expression to modulate physiological processes may provide insights into disease mechanisms and highlight potential therapeutic targets.

Steroid receptor coactivator (SRC)-1 belongs to a family of coactivators (SRC-1, -2, and -3) that mediate NR-dependent or TF-dependent transcription[2]. Global deletion of SRC-1 in mice leads to obesity[3]; however, to date, the molecular mechanisms involved are incompletely understood. SRC-1 is abundantly expressed in the hypothalamus, including neurons within the arcuate nucleus of the hypothalamus (ARH)[4], which play a key role in mediating the weight-reducing effects of the adipocyte-derived hormone leptin[5,6]. Leptin is a signal of nutrient deprivation, with a fall in leptin levels triggering a set of responses that seek to restore energy homeostasis by increasing food intake and decreasing energy expenditure[7]. In the fed state, an increase in leptin levels leads to the activation of neurons expressing the anorectic peptide Pro-opiomelanocortin (POMC) leading to a reduction in food intake[8]. Specifically, leptin binding to its receptor phosphorylates the transcription factor STAT3 which dimerizes and translocates to the nucleus where it stimulates the expression of POMC[9–11]. Leptin-induced STAT3 activation also stimulates expression of Socs3 (suppressor of cytokine signaling-3) which acts to inhibits leptin signaling[12,13].

In this study, we sought to investigate the central mechanisms by which SRC-1 modulates energy homeostasis. SRC family members bind to STAT transcription factors in cells[14]. Thus, we first examined the effects of SRC-1 on STAT3 transcriptional activity and Pomc expression. We then characterized metabolic phenotypes in mice lacking SRC-1 in Pomc neurons and explored the underlying mechanisms. Additionally, we examined the potential functional consequences of rare human variants in SRC-1 identified in severe childhood-onset obesity. Finally, we generated a knock-in mouse model of the most severe loss of function human SRC-1 variant and characterized the metabolic consequences of these mutant mice.

## Results

**SRC-1 interacts with pSTAT3 to stimulate Pomc expression.** We found that global SRC-1-KO mice[15] had lower Pomc but normal Socs3 mRNA levels in the hypothalamus compared to control littermates (Fig. 1a). Using Chromatin-immunoprecipitation (ChIP) assays, we found that leptin-stimulated pSTAT3 binding to Pomc promoters was decreased

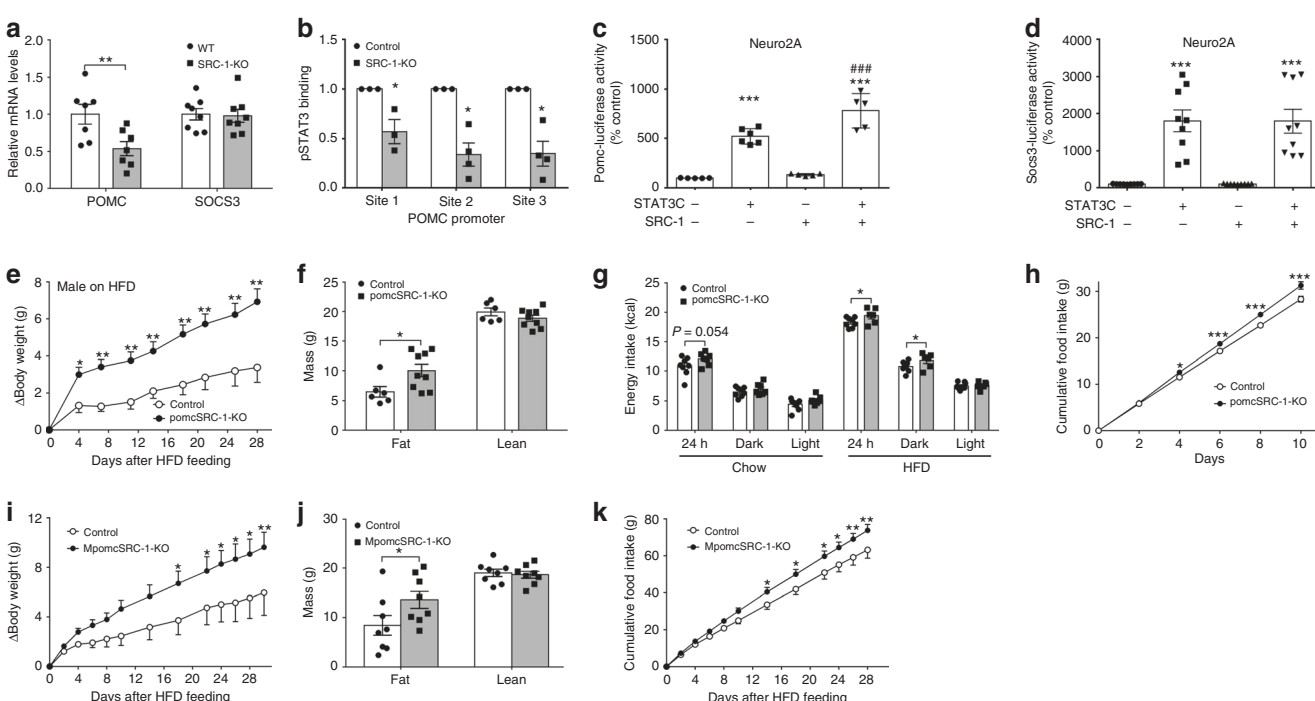

**Fig. 1** SRC-1 potentiates STAT3-induced Pomc expression. Numbers of mice/repeats in each group are indicated; data are presented as mean ± SEM and compared using T-tests or two-way ANOVA followed by post hoc Sidak tests (#). **a** Pomc and Socs3 mRNA levels in hypothalami from 16-week old SRC-1-KO and WT control littermates (n = 7/8); **P < 0.01. **b** ChIP assays detecting pSTAT3 binding on Pomc promoters in hypothalami from male SRC-1-KO and control littermates 30 min after leptin injections (5 mg/kg, i.p.): site 1, −998 to −989; site 2, −361 to −353; site 3, −76 to −68 upstream of Pomc (n = 3/4); *P < 0.05. **c, d** Effects of overexpressed constitutively active STAT3 and SRC-1 on Pomc- (**c**) or Socs3-luciferase activity (**d**) in Neuro2A cells (n = 5–9 independent experiments). ***P < 0.001 vs. empty vectors; ###P < 0.001 vs. STAT3 alone (#). **e** Change (Δ) in body weight after male control and pomcSRC-1-KO mice were switched onto a HFD at day 97 (n = 6/9); *P < 0.05 and **P < 0.01 (#). **f** Fat mass and lean mass measured 28 days after HFD feeding (n = 6/9); *P < 0.05. **g** Energy intake measured by CLAMS chambers in 12-week old male mice matched for body weight, lean mass, and fat mass. Mice were subjected to a 2-day-chow–2-day-HFD protocol, and chow was replaced by HFD before the onset of dark cycle on day 3. Energy intake was averaged for 2-day chow feeding period and for 2-day HFD feeding period (n = 7/8); *P < 0.05. **h** Cumulative HFD intake measured in 12-week old male mice singly housed in home cages (n = 10/14); *P < 0.05 (#). **i** Change in body weight after control and MpomcSRC-1-KO mice were switched on a HFD at the age of day 84 (n = 8); *P < 0.05 (#). **j** Fat mass and lean mass measured 30 days after HFD feeding (n = 8); *P < 0.05. **k** Cumulative HFD intake measured in 12-week old male mice (n = 6/7); *P < 0.05 (#), **P < 0.01. Source data are provided as Source Data Fig. 1

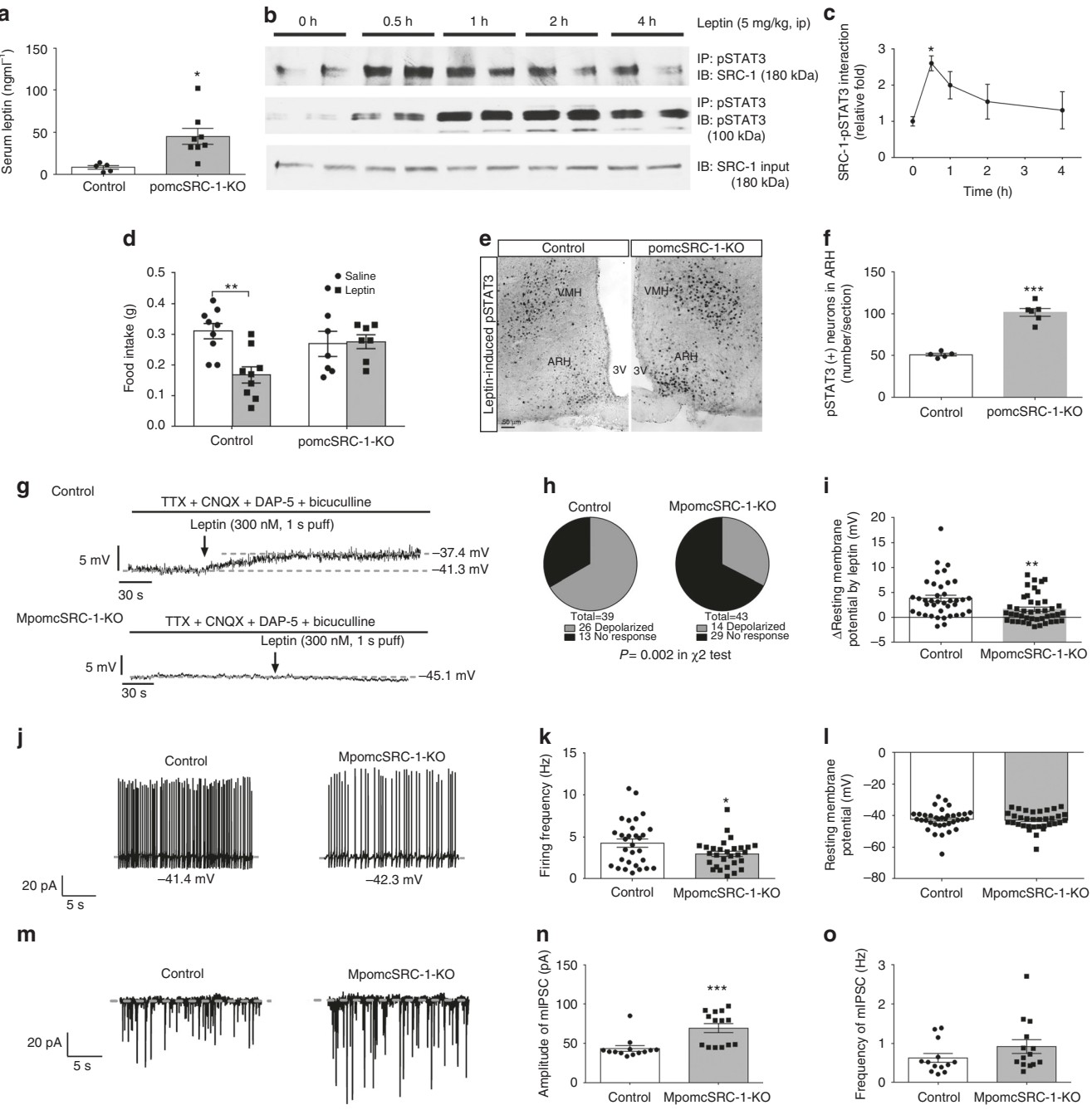

**Fig. 2** SRC-1 mediates leptin signaling. Numbers of mice/experiments/neurons are indicated; data are presented as mean ± SEM and compared using T-tests or one- or two-way ANOVA followed by post hoc Sidak tests (#). **a** Serum leptin levels 42 days after HFD feeding ($n = 5/8$); *$P < 0.05$. **b** Time course of hypothalamic SRC-1-pSTAT3 interaction in C57Bl6 wild type mice that received i.p. injections of leptin (5 mg/kg). **c** Quantification of the hypothalamic SRC-1-pSTAT3 interaction. *$P < 0.05$ (#). **d** Two-hour fasted mice (12 weeks of age) received i.p. injections of saline or leptin (5 mg/kg) 15 min prior to refeeding and food intake was recorded for 1 h afterwards ($n = 7/9$); **$P < 0.01$ (#). **e** Representative pSTAT3 immunohistochemical staining in the ARH and VMH of control and pomcSRC-1-KO mice receiving a single bolus i.p. injection of leptin (0.5 mg/kg, 90 min). Scale bar = 50 μm. 3V the 3rd ventricle, ARH arcuate nucleus, VMH ventromedial hypothalamic nucleus. **f** Quantification of pSTAT3 (+) neurons in the ARH ($n = 5$); ***$P < 0.001$. **g** Representative traces of leptin-induced depolarization, in the presence of TTX, CNQX, DAP-5, and bicuculline, in mature Pomc neurons from control mice vs. from MpomcSRC-1-KO mice after 1-week HFD feeding. **h** Responsive ratio (depolarization is defined as >2 mV elevations in resting membrane potential) ($n = 39/43$); $P = 0.002$ in $\chi^2$ tests. **i** Quantification of leptin-induced depolarization in two groups ($n = 39/43$); **$P < 0.01$. **j** Representative traces of action potentials in untreated mature Pomc neurons from control mice vs. from MpomcSRC-1-KO mice. **k, l** Quantification of firing frequency (**k**) and resting membrane potential (**l**) in two groups ($n = 29–36$); *$P < 0.05$. **m** Representative traces of mIPSC in untreated mature Pomc neurons from control mice vs. from MpomcSRC-1-KO mice. **n, o** Quantification of amplitude (**n**) and frequency (**o**) of mIPSC in two groups ($n = 13/14$); ***$P < 0.001$. Source data are provided as Source Data Fig. 2

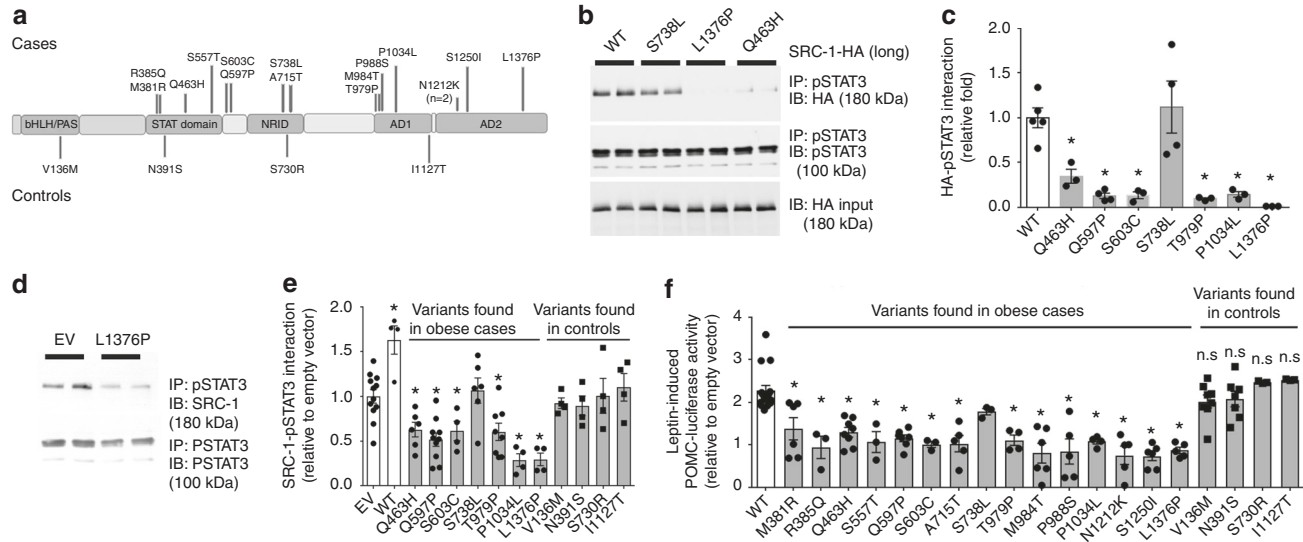

**Fig. 3** Missense variants in SRC-1 disrupt leptin signaling. Numbers of experiments are indicated; data are presented as mean ± SEM and compared using one-way ANOVA followed by post hoc Sidak tests unless mentioned otherwise. **a** Rare variants identified in individuals with severe early onset obesity (above) and in controls (below). **b**, **c** HEK293 cells were co-transfected with leptin receptor vector and human STAT3 vector. Cells were treated with leptin (200 ng/ml, 15 min) to induce phosphorylation of STAT3. pSTAT3 was pulled down using anti-pSTAT3 sepharose beads; beads were then aliquoted equally and incubated with the same amount of the long isoform of human SRC-1-HA (WT/mutant) and interactions between the pSTAT3 and SRC-1 were determined by CoIP experiments using anti-pSTAT3 and anti-HA antibodies. **b** Representative blots showing interactions between pSTAT3 and SRC-1 (WT/mutant), and inputs of pSTAT3 and SRC-1-HA. **c** Quantification for WT and SRC-1 mutants. Comparative folds were calculated as the ratios of HA blots and HA inputs ($n = 3$–5); *$P < 0.05$. **d**, **e** SRC-1 mutants inhibit the interaction between STAT3 and WT SRC-1. HEK293 cells were co-transfected with leptin receptor vector, STAT3 vector, and mutant SRC-1 vector (or empty vector). Cells were treated with leptin (200 ng/ml, 15 min) to induce phosphorylation of STAT3 and interactions between pSTAT3 and total SRC-1 were determined by CoIP experiments using anti-pSTAT3 and anti-SRC-1 antibodies. **d** Representative blots showing interactions between pSTAT3 and SRC-1 variants found in obese cases and inputs of pSTAT3. **e** Quantification. Comparative folds were calculated as the ratios of SRC-1-pSTAT3 interaction blots and pSTAT3 inputs ($n = 4$–12); *$P < 0.05$. **f** SRC-1 variants impair POMC expression. Neuro2A cells were co-transfected with leptin receptor vector, SRC-1 (WT or mutant) and a POMC luciferase expression reporter construct. Cells were stimulated with 200 ng/ml leptin for 15 min and then incubated for 6 h, following which luminescence was measured. Results were normalized to empty vector-induced expression ($n = 3$–16); *$P < 0.05$. Source data are provided as Source Data Fig. 3

in the hypothalamus of *SRC-1-KO* mice compared to control mice (Fig. 1b). In keeping with these findings, *SRC-1* over-expression potentiated STAT3-induced *Pomc* transcription but had no effect on *Socs3* transcription in Neuro2A cells and HEK293 cells (Fig. 1c, d; Supplementary Figure 1a-b). Similar effects of *SRC-1* were observed in *SRC-1-KO* MEFs cells, although STAT3 alone could stimulate *Pomc* expression in these cells devoid of endogenous SRC-1 (Supplementary Figure 1c-d). These results indicate that SRC-1, while not required for STAT3 tran-scriptional activity, can facilitate STAT3-induced *Pomc* expression.

**SRC-1 in Pomc neurons regulates energy homeostasis.** To test whether SRC-1 in Pomc neurons plays a functionally significant role in energy homeostasis, we crossed *SRC-1^lox/lox^* mice with *Pomc-Cre* mice to generate mice lacking *SRC-1* selectively in *Pomc* lineage cells (pomcSRC-1-KO, Supplementary Figure 1e). On a standard chow diet, the body weight of male pomcSRC-1-KO mice was comparable to control littermates (*SRC-1^lox/lox^*) (Supplementary Figure 1f), whilst female pomcSRC-1-KO mice showed significant weight gain (Supplementary Figure 1g). This sexual dimorphism may be explained by our earlier observations that global *SRC-1* deficiency blunts the weight-reducing effects of estrogen[4]. On a high fat diet (HFD), male pomcSRC-1-KO mice gained significantly more weight compared to control littermates (Fig. 1e) due to an increase in fat mass (Fig. 1f). In weight-matched mice, we observed a significant increase in HFD intake in pomcSRC-1-KO mice vs. controls (Fig. 1g, h); measurements

of energy expenditure were comparable (Supplementary Figure 1h–j).

A caveat of the regular *Pomc-Cre* mouse line is that, during the early development, Cre recombinase is transiently expressed in a broader population of neurons and some of these *Pomc* lineage cells mature into orexigenic Npy/Agrp neurons with opposing effects on food intake[16]. To address this concern, we crossed a *Pomc-CreER* transgene[17] onto the *SRC-1^lox/lox^* mouse allele. Tamoxifen induction at 9 weeks of age resulted in the deletion of SRC-1 in mature *Pomc* neurons (MpomcSRC-1-KO; Supple-mentary Figure 1k-l). When fed with a HFD, MpomcSRC-1-KO mice displayed increased weight gain and fat mass, associated with increased food intake compared to control littermates (Fig. 1i–k), which recapitulated the phenotypes observed in pomcSRC-1-KO mice. Collectively, these results indicate that SRC-1 in mature *Pomc* neurons is required to defend against diet-induced obesity.

**SRC-1 in Pomc neurons is required for the anorectic effects of leptin.** Several studies have shown that STAT3 signaling is a mediator of leptin's effects on body weight[10,18]. In HFD-fed *pomcSRC-1-KO* mice, we observed a 5–6-fold increase in circulating leptin levels in HFD-fed *pomcSRC-1-KO* mice (Fig. 2a), whilst adiposity only increased 2-fold (Fig. 1f). Thus, we hypo-thesized that SRC-1 is downstream of leptin action and loss of SRC-1 in *Pomc* neurons may impair leptin signaling. Supporting this possibility, we found that intra-peritoneal administration of leptin to control mice rapidly increased the hypothalamic SRC-1-

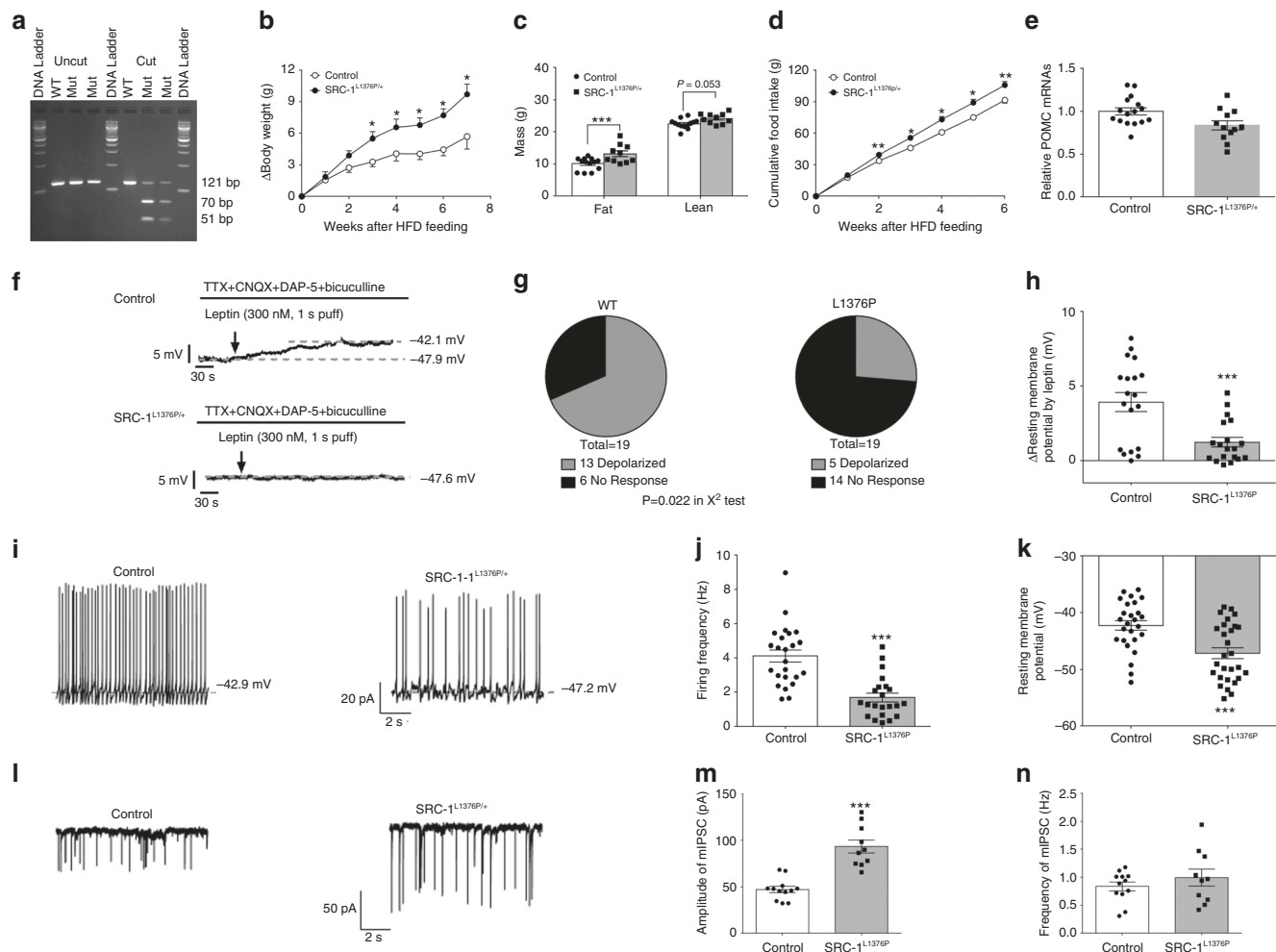

**Fig. 4** $SRC\text{-}1^{L1376P/+}$ mice are obese. Numbers of mice in each group are indicated; data are presented as mean ± SEM and compared using $T$-tests or two-way ANOVA followed by post hoc Sidak tests (#). **a** The PCR products (121 bp) around the L1376 were amplified from genomic DNA extracts of a WT and two $SRC\text{-}1^{L1376P/+}$ mutant mice and incubated with or without Sau3AI. Control reaction (WT) resulted in a single large fragment (121 bp) and DNAs from the two $SRC\text{-}1^{L1376P/+}$ mutant mice were cut into two fragments (70 and 51 bp) as expected. **b** Change in body weight after male control and SRC-1$^{L1376P/+}$ mice were fed on a HFD ($n = 5/6$); *$P < 0.05$ (#). **c** Fat mass and lean mass measured 7 weeks after HFD feeding ($n = 5/6$); ***$P < 0.001$. **d** Cumulative HFD intake measured ($n = 5/6$); *$P < 0.05$ or **$P < 0.01$ (#). **e** Pomc mRNA levels in hypothalami from 20-week old HFD-fed male control and SRC-1$^{L1376P/+}$ mice ($n = 12/16$); *$P < 0.05$. **f** Representative traces of leptin-induced depolarization, in the presence of TTX, CNQX, DAP-5, and bicuculline, in Pomc neurons from control mice vs. from $SRC\text{-}1^{L1376P/+}$ mice after 1-week HFD feeding. **g** Responsive ratio (depolarization is defined as >2 mV elevations in resting membrane potential) ($n = 19$); $P = 0.022$ in $\chi^2$ tests. **h** Quantification of leptin-induced depolarization in two groups ($n = 19$); ***$P < 0.001$. **i** Representative traces of action potentials in untreated Pomc neurons from control mice vs. from $SRC\text{-}1^{L1376P/+}$ mice. **j, k** Quantification of firing frequency (**j**) and resting membrane potential (**k**) in two groups ($n = 22\text{--}28$); ***$P < 0.001$. **l** Representative traces of mIPSC in untreated Pomc neurons from control mice vs. from $SRC\text{-}1^{L1376P/+}$ mice. **m, n** Quantification of amplitude (**m**) and frequency (**n**) of mIPSC in two groups ($n = 10/12$); ***$P < 0.001$. Source data are provided as Source Data Fig. 4

pSTAT3 interaction (Fig. 2b, c). Leptin administration significantly reduced 1-hour (1h) food intake in control mice but not in pomcSRC-1-KO mice (Fig. 2d), despite increased leptin-induced pSTAT3 in the arcuate nucleus (Fig. 2e, f). These results suggest that the SRC-1-pSTAT3 interaction is downstream of leptin-STAT3 signaling, and contributes to the acute anorectic effects of leptin. Notably, the effects of leptin on 4 and 24 h food intake were not significantly altered in pomcSRC-1-KO mice (Supplementary Figure 2a-b), presumably because the anorectic effects of leptin after the first hour are mediated by other leptin-responsive neurons or other signaling pathways[19,20].

Leptin also depolarizes a subset of Pomc neurons to exert its anorectic effects[8], although recent fiber photometry studies failed to detect acute effects of leptin on calcium dynamics in Pomc neurons[21]. Thus, we examined leptin-induced depolarization in

TOMATO-labeled mature Pomc neurons from $MpomcSRC\text{-}1\text{-}KO$ mice and tamoxifen-treated controls after 1-week HFD feeding. We recorded leptin-induced changes in resting membrane potential (RM) in the presence of tetrodotoxin (TTX), which blocks action potentials, and a mixture of fast synaptic inhibitors which block the majority of presynaptic inputs. We found that 26/39 (67%) of Pomc neurons from control mice were depolarized (>2 mV elevations in RM) by leptin (Fig. 2g, h). In contrast, only 14/43 (33%) of Pomc neurons from $MpomcSRC\text{-}1\text{-}KO$ mice were depolarized by leptin ($P = 0.002$) and the amplitude of leptin-induced depolarization was significantly reduced in these Pomc neurons (Fig. 2g–i). Interestingly, in the absence of TTX and synaptic inhibitors, leptin-induced depolarization and increases in firing frequency were comparable between the two groups (Supplementary Figure 2c–f), suggesting

that indirect effects of leptin through presynaptic terminals[22,23] were not affected by the loss of SRC-1 in Pomc neurons. Notably, the baseline firing frequency was significantly decreased in mature Pomc neurons from MpomcSRC-1-KO mice compared to those from control mice, whereas the baseline RM remained unchanged (Fig. 2j–l). We found that the amplitude, but not the frequency, of miniature inhibitory postsynaptic currents (mIPSC) was significantly higher in mature Pomc neurons from MpomcSRC-1-KO mice than those from control mice (Fig. 2m–o). The frequency of mIPSC is thought to reflect presynaptic events (e.g., GABA release), while mIPSC amplitude is largely determined by the responsiveness of postsynaptic neurons. Thus we suggest that SRC-1 also regulates the responsiveness of Pomc neurons to GABA-ergic inputs via a leptin-independent mechanism.

**Rare SRC-1 variants found in obese humans impairs SRC-1 functions.** We next investigated the potential role of SRC-1 in humans by interrogating exome sequencing and targeted resequencing data on 2548 European ancestry individuals with severe, early-onset obesity (mean body mass index [BMI] standard deviation score = 3; age of onset < 10 years) and 1117 ancestry-matched controls[24]. Eleven rare heterozygous variants in *SRC-1* were identified; another 8 variants were identified in an earlier data release (total $n = 19$). Fifteen *SRC-1* variants were identified only in obese cases (N1212K was found in two unrelated obese individuals); the other 4 variants were found in controls (Fig. 3a). Compared to WT SRC-1, six of seven randomly selected SRC-1 mutants found in obese cases (except for S738L) were significantly impaired in their interaction with pSTAT3 in leptin-treated HEK293 cells (Fig. 3b, c, Supplementary Figure 3a–c). To test whether heterozygous *SRC-1* variants exerted a dominant negative effect to inhibit the interaction between WT SRC-1 and pSTAT3, we overexpressed SRC-1 mutants in HEK293 cells which endogenously express SRC-1. After leptin treatment, an anti-pSTAT3 antibody was used to pull down the immunocomplex from cell lysates, followed by immunoblotting with an anti-SRC-1 antibody to examine the interaction between pSTAT3 and total SRC-1. Overexpression of SRC-1 mutants found in obese cases (6 of 7 tested mutants) significantly decreased the interaction between pSTAT3 and the total SRC-1, suggesting that these SRC-1 mutants can impair the ability of WT SRC-1 to interact with pSTAT3 (Fig. 3d, e and Supplementary Figure 3d-e). This dominant negative effect was not seen when testing the 4 mutants found in controls (Fig. 3e and Supplementary Figure 3d). We used a POMC-luciferase reporter assay to examine the effects of leptin on *Pomc* expression. We found that WT SRC-1 significantly enhanced leptin-induced Pomc-luciferase reporter activity, but co-expression of a dominant negative form of STAT3 abolished this effect (Supplementary Figure 3f-g), suggesting that the interaction with STAT3 is required for the observed effects of SRC-1 on *Pomc* transcription. Fourteen of fifteen SRC-1 mutants found in severely obese cases (except for S738L) significantly impaired leptin-induced *Pomc* expression, whereas the 4 control mutants exhibited WT-like responses in this assay (Fig. 3f). Interactions with estrogen receptor-α, vitamin D receptor, glucocorticoid receptor, thyroid hormone receptor-β, and peroxisome proliferator-activated receptor γ (PPARγ) were comparable to those seen for WT SRC-1 (Supplementary Figure 4) in co-immunoprecipitation assays.

**A mouse model of the human SRC-1 variant L1376P is obese.** To directly test whether rare human SRC-1 variants contribute to Pomc neuron function and/or energy homeostasis, we generated a knock-in mouse model of a human variant which results in a severe loss of function in cells, SRC-1$^{L1376P}$ (Fig. 4a). Heterozygous mutant mice (SRC-1$^{L1376P/+}$) fed a HFD exhibited increased weight gain, adiposity and food intake, associated with reduced Pomc mRNA levels compared to WT controls (Fig. 4b–e). We recorded leptin-induced depolarization in Pomc neurons in control vs. SRC-1$^{L1376P/+}$ mice 1 week after HFD feeding. In control mice, 13/19 (68%) Pomc neurons were depolarized by leptin, whilst only 5/18 (26%) Pomc neurons from SRC-1$^{L1376P/+}$ mice were depolarized by leptin ($P = 0.022$) and the amplitude of leptin-induced depolarization was significantly reduced in these Pomc neurons (Fig. 4f–h). Baseline firing frequency and resting membrane potential were both significantly decreased in Pomc neurons from SRC-1$^{L1376P/+}$ mice compared to those from control mice (Fig. 4i–k). Further, the amplitude, but not the frequency, of the mIPSC was significantly higher in Pomc neurons from SRC-1$^{L1376P/+}$ mice than those from control mice (Fig. 4l–n). Thus, these data indicate that the SRC-1$^{L1376P}$ variant causes obesity in mice, associated with decreased Pomc expression and decreased Pomc neuron excitability through both leptin-dependent and independent mechanisms.

## Discussion

In this study, we demonstrated that in the hypothalamus, the coactivator SRC-1 modulates the ability of leptin to regulate the expression of the anorectic peptide POMC by directly interacting with phosphorylated STAT3, a known product of leptin-receptor activation. In mice, disruption of SRC-1 in Pomc neurons led to increased food intake, weight gain on a HFD and impaired the acute anorectic response to leptin administration demonstrating the physiological relevance of this molecular interaction. The modest degree of obesity in these mice was comparable to that seen with inactivation of STAT3 in Pomc neurons[9] and studies demonstrating that direct leptin action on Pomc neurons accounts for a proportion of leptin's effects on body weight[19,20,25,26]. The obesity seen in SRC-1 deletion or mutant mice was less severe than that see in mice deficient in Pomc[27] or melanocortin 4 receptor[28] in keeping with SRC-1's role as a modulator of Pomc expression. Additionally, leptin-responsive Agrp neurons have been shown to play a major role in energy homeostasis[20].

We identified 15 rare heterozygous variants in *SRC-1* in 16 severely obese individuals and 4 rare variants in controls. Notably, there are several low frequency and many rare variants in this gene in publically available databases (http://gnomad.broadinstitute.org/). Some of these low frequency variants have been shown to have functional consequences, for example, P1272S (MAF: 3.16% in cases, 3.45% in controls; 1.66% in gnomAD) disrupts a putative glycogen synthase 3 (GSK3)β phosphorylation site and has been shown to exhibit reduced ability to co-activate Estrogen Receptor in multiple cell lines[29]. Genetic studies in larger numbers of cases and controls with functional studies of all variants identified will be needed to establish whether variants that result in a loss of function when tested in cells are more likely to be found in severely obese individuals than in controls. In this study, the variants found in obese individuals, but not those found in controls, were associated with impaired interaction with pSTAT3 and reduced POMC reporter activity in cells, predominantly through a dominant negative effect. Given the challenges associated with studying such rare variants, and to directly test whether rare human SRC-1 variants contribute to Pomc neuron function and/or energy homeostasis, we generated a knock-in mouse model of a human variant which results in a severe loss of function human SRC-1 variant supports the

potential importance of the mechanism identified here in humans.

Recent evidence indicates that loss of leptin receptors in Pomc neurons does not affect body weight in chow-fed mice[19,20]. In line with these reports, we show that loss of SRC-1 in Pomc neurons produced minor effects on energy balance in chow-fed male mice. These suggest that the physiological consequences of disrupting this interaction in normal weight animals are small and/or may be compensated for by increased signaling through non-POMC expressing leptin-responsive neurons[30] and/or signaling via phosphoinositide-3-kinase (PI3K)[31], mTOR/S6K[32] and/or AMPK pathways[33,34]. We showed that SRC-1 deletion in Pomc neurons attenuated the acute anorectic response (1 h) to leptin but not the late phase (4–24 h). Cumulatively, these findings indicate that leptin-mediated POMC expression (modulated by the SRC-1-pSTAT3 interaction) primarily contributes to the acute anorectic response to leptin. In keeping with this finding, we demonstrated that the hypothalamic SRC-1-pSTAT3 interaction was enhanced by leptin. Consumption of HFD leads to sustained positive energy balance and an increase in leptin levels. The resulting increase in pSTAT3 would be expected to stimulate POMC expression and reduce food intake, a response that we have shown is modulated by the interaction between pSTAT3 and SRC-1. We suggest that in the absence of functional SRC-1, pSTAT3 is less effective at stimulating POMC expression, which manifests as a relative increase in food intake and weight gain when mice are challenged with HFD. In this way, we conclude that SRC-1 acts as a positive regulator of leptin sensitivity in hypothalamic Pomc neurons.

Our findings suggest that SRC-1 facilitates but is not required for pSTAT3 to regulate Pomc expression and that this effect is target-specific as SRC-1 does not modulate the ability of pSTAT3 to regulate Socs3. The mechanisms underlying such specificity remain unclear at present. The molecular interaction between SRC-1 and pSTAT3 enhances pSTAT3-mediated transcriptional activity, presumably by stabilizing pSTAT3 binding to the POMC promoter, although we cannot exclude the possibility that recruitment of other co-coactivators or histone acetyltransferase activity of SRC-1 also may be involved[35]. Further studies of the molecular mechanisms that modulate leptin signaling are emerging[36–41]. For example, Chen et al showed that the nuclear receptor Nur77 facilitates STAT3 acetylation by recruiting acetylase p300 and disassociating deacetylase histone deacetylase 1 (HDAC1) to enhance the transcriptional activity of STAT3[42]. In findings that parallel our studies, they showed that Nur77 deficiency reduced the expression of Pomc in the hypothalamus and attenuated the response to leptin in mice fed on a HFD[42].

Transcriptional coactivators such as SRC-1 facilitate the signaling mediated by multiple NRs and/or TFs factors[2]. Several NRs/TFs have been shown to affect energy homeostasis through their actions in the brain[43], including FoxO1[44–47], ERα[48,49], PPARγ[50,51], and THR[52] and thus could contribute to the body weight phenotype seen with SRC-1 disruption in mice and loss of function variants in humans. In addition to the central actions of SRC-1 on energy homeostasis, SRC-1 is expressed in brown adipose tissue, where it appears to compete with SRC-2 to interact with the PPARγ-PGC1α complex. Picard et al showed that SRC-1-KO mice had reduced rectal temperatures upon cold exposure and reduced oxygen consumption although they did not quantify food intake in this study[3]. Notably, we did not observe any changes in energy expenditure in mice lacking SRC-1 in Pomc neurons, consistent with the notion that SRC-1 in other tissues may also contribute to the regulation of energy expenditure[3]. Whilst we found that SRC-1 variants detected in obese patients did not affect the interactions with a number of NRs, these results do not exclude the potential impact of SRC-1 variants on the signaling of these NRs which need to be explored in

more detail using tissue-specific conditional knockout mouse models.

Targeting specific coactivator-mediated interactions has emerged as a potential therapeutic strategy to enhance signaling in some tissues while inhibiting signaling in others[53,54]. For example, Selective Estrogen Receptor Modulators (SERMs) are effective in modulating the growth of hormone-responsive tumors (e.g., Tamoxifen in breast cancer) by impacting on coactivator stability and activity[55]. As such, compounds that target the interaction between SRC-1 and STAT3 at specific sites may potentially be used to modulate (i.e., enhance) leptin signaling. Could this approach be efficacious in the treatment of obesity? Studies in mice and humans have consistently demonstrated that leptin sensitivity is greatest in those with no/very low endogenous circulating leptin levels[56,57]. Whether enhancing leptin sensitivity in the context of common obesity, which is associated with elevated leptin levels, may be clinically beneficial, is the subject of much debate[18,58–60]. The finding that some compounds (e.g., the amylin derivative pramlintide) can augment the effects of leptin[61,62], suggests that it may be possible to increase the sensitivity of some individuals to therapeutic leptin administration and that this approach may lead to weight loss. These observations and our findings on SRC-1 suggest that pharmacological approaches based on the modulation of leptin sensitivity could represent a potential therapeutic strategy for the treatment of obesity-associated metabolic disease.

## Methods

**Contact for reagent and resource sharing**. Further information and requests for resources and reagents should be directed to and will be fulfilled by Yong Xu (yongx@bcm.edu) and Sadaf Farooqi (isf20@cam.ac.uk).

**Experimental model and subject details**. Mice: We crossed regular Pomc-Cre transgenic mice[26] and SRC-1[lox/lox] mice[63]. This cross produced pomcSRC-1-KO mice (those that are homozygous for SRC-1[lox/lox] and also carry the Pomc-Cre transgene) and control mice (those that are homozygous for SRC-1[lox/lox] but do not carry the Pomc-Cre transgene). These littermates were used to characterize the metabolic profile.

In addition, we also crossed inducible Pomc-CreER mice[17] with SRC-1[lox/lox] mice to generate MpomcSRC-1-KO mice (those that are homozygous for SRC-1[lox/lox] and also carry the Pomc-CreER transgene) and control mice (those that are homozygous for SRC-1[lox/lox] but do not carry the POMC-CreER transgene). Both these mice received tamoxifen injections (0.2 mg/g, i.p., twice at 9 weeks of age). These littermates were used to characterize the metabolic profile. For electrophysiological recordings, we crossed the inducible Pomc-CreER and the Rosa26-tdTOMATO mouse alleles onto SRC-1[lox/lox] mice, to produce MpomcSRC-1-KO mice with mature Pomc neurons labeled by TOMATO; as controls, we crossed inducible Pomc-CreER mice and Rosa26-tdTOMATO mice to generate Pomc-CreER/Rosa26-tdTOAMTO mice. In parallel, we also crossed the Npy-GFP mouse allele[23] and the Rosa26-tdTOMATO allele onto inducible Pomc-CreER mice. This cross produced Pomc-CreER/Rosa26-tdTOAMTO/Npy-GFP mice, which were subjected to histology validation for the inducible Pomc-CreER mice.

To generate the SRC-1[L1376P/+] knock-in mice, a single-guide RNA (sgRNA) sequence was selected overlap amino acid residue L1382 (equivalent to human L1376) in SRC-1 (sgRNA 5′-CATCTGCGTCTGTTTTGAGAagg chr12:4253665-4253687; GRCm38/mm10) using the CRISPR Design Tool (Ran et al. 2013). A DNA templates for in vitro transcription of the sgRNA was produced using overlapping oligonucleotides in a high-fidelity PCR reaction[64], and sgRNA was transcribed using the MEGAshortscript T7 kit (ThermoFisher, Waltham, MA). Cas9 mRNA was purchased from ThermoFisher. The donor DNA template to introduce the L1382P point mutation, as well as a silent mutation D1381D to introduce a novel restriction site for Sau3AI, was purchased as an Ultramer from IDT (Coralville, IA). The sequence of ssODN is as follows (complementary to non-target strand): 5′ TGAAAATCTG CTCTTTTGTT TATCCTTAAT AGATGAATG A TCCAGCACTG AGACACACAG GCCTCTACTG CAACCAGCTC TCGTCCA CTG ATCCCCTCAA AACAGACGCA GATGGAAACC AGGTCAGTAA GAAA, where the mutations are in bold. The mutations introduced in the donor sequence disrupt base 20 of the sgRNA and the PAM site to prevent additional mutagenesis. The BCM Genetically Engineered Mouse (GEM) Core microinjected Cas9 mRNA (100 ng/µl), Ultramer ssDNA (100 ng/µl), and sgRNA (20 ng/µl) into the cytoplasm of 200 pronuclear stage C57Bl/6J embryos. Cytoplasmic injections were performed using a microinjection needle (1 mm outer and 0.75 mm inner) with a tip diameter of 0.25–0.5 µm, an Eppendorf Femto Jet 4i to set pressure and time to control injection volume (0.5–1 pl per embryo). Injections were performed

under a 200–400× magnification with Hoffman modulation contrast for visualizations. Founder animals (F₀) were identified by PCR-based restriction digestion to detect the CRISPR generated point mutations in SRC-1. PCR product was first amplified with the primer pairs: 5′-CCTCACTT GTGGCAATGTGA and 5′-TCGTGGCAGTTCTGTAGTCAC; and then amplified with 2nd pairs: 5′-CACTGAGACACACAGGCCTC and 5′-ATCGAATCTG CCAGCTCTGC. The 121 bp PCR products were then digested with Sau3AI. 70 and 51 bp products after digest could be detected only for the mutated SRC-1 PCR products. Three independent lines were sequenced for the further confirmation of the point mutation. One of these lines was crossed to C57Bl6j to produce cohorts comprised of *SRC-1*[L1376P/+] and wild-type control mice. In some breedings, the *Pomc-CreER-tdTOAMTO* alleles were introduced to allow specific labeling of Pomc neurons.

In parallel, we crossed heterozygous *SRC-1-KO* mice[65] to heterozygous *SRC-1-KO* mice to produce homozygous *SRC-1-KO* and wild-type littermates. All the breeders have been backcrossed to C57Bl6 background for more than 12 generations. In addition, some C57Bl6 mice were purchased from the mouse facility of Baylor College of Medicine.

Care of all animals and procedures were approved by the Institutional Animal Care and Use Committee (IACUC) of Baylor College of Medicine Animal Facility, and all experimental procedures in animals complied with all relevant ethical regulations. Mice were housed in a temperature-controlled environment in groups of 2–5 at 22–24 °C using a 12 h light/12 h dark cycle. Some cohorts were singly housed to measure food intake. The mice were fed either standard chow (6.5% fat, #2920, Harlan-Teklad, Madison, WI), or a 60% HFD (60% fat, #D12492, Research Diets). Water was provided ad libitum.

**Studies in mice.** *Validation of genomic deletion of SRC-1 in Pomc cells*: Control mice, pomcSRC-1-KO mice or MpomcSRC-1-KO mice (after tamoxifen inductions) were anesthetized with inhaled isoflurane, and sacrificed. Various tissues, as detailed in the figures, were collected. Genomic DNAs were extracted using the REDExtract-N-Amp Tissue PCR Kit (#XNATS; Sigma-Aldrich, St Louis, MO), followed by PCR amplification of the floxed or recombined alleles. We used primers: forward-CAGTAAGGAATAGCAGATGTC and reverse-TGGCATCTATAACCAAATGTGTA TCA to detect the wild-type allele (a 560 bp band) and the floxed SRC-1 allele (a 630 bp band); and combined the reverse primer (mentioned above) with another forward primer: GTCGTACCATC-TATGCCTCCTATAT to detect the recombined SRC-1 allele (a 320 bp band).

*Histology*: To validate specificity of the inducible *Pomc-CreER* transgene, *Pomc-CreER/Rosa26-tdTOAMTO/Npy-GFP* mice received tamoxifen injections (0.2 mg/g, i.p., twice) at 9 weeks of age, and then were perfused 1 week later. Brain sections were cut at 25 μm (1:5 series) and subjected to direct visualization of GFP and TOMATO signals using a Leica DM5500 fluorescence microscope with OptiGrid structured illumination configuration.

To examine the effects of leptin on STAT3 phosphorylation in vivo, control and pomcSRC-1-KO mice (5 or 6 per group) were fasted overnight and then received a single bolus injection of saline or leptin (0.5 mg/kg, i.p.). Ninety minutes after the bolus injections, mice were anesthetized with inhaled isofluorane, and quickly perfused with 10% formalin, and brain sections were cut at 25 μm. The brain sections were pretreated (1% H₂O₂, 1% NaOH, 0.3% glycine, 0.03% SDS), blocked (3% goat-anti-rabbit serum for 1 h), incubated with rabbit anti-pSTAT3 antibody (1:2000; #9145, Cell Signaling) on shaker at room temperature for 24 h and then put in 4 °C for 48 h, followed by biotinylated anti-rabbit secondary antibody (1:1000; Vector) for 2 h. Sections were then incubated in the avidin–biotin complex (1:500, ABC; Vector Elite Kit) and incubated in 0.04% 3,3′-diaminobenzidine and 0.01% hydrogen peroxide. After dehydration through graded ethanol, the slides were then immersed in xylene and cover-slipped. Images were analyzed using a brightfield Leica microscope. The numbers of pSTAT3-positive neurons in the ARH were counted by blinded investigators. For each mouse, pSTAT3-positive neurons were counted in 3–5 consecutive brain sections containing ARH, and the average was treated as the data value for that mouse. Five or six mice were included in each group for statistical analyses.

*Body weight study*: pomcSRC-1-KO mice and their control littermates were weaned at week 4. These mice were group housed and maintained on the standard chow (6.5% fat, #2920, Harlan-Teklad). At the age of day 97, mice were switched to the HFD (60% fat, #D12492, Research Diets) for 6 weeks. Body weight was measured every 4 days since weaning. Body composition was determined using quantitative magnetic resonance (QMR) on 28 days after HFD feeding. On day 42 after HFD feeding, the mice were deeply anesthetized with inhaled isoflurane and sacrificed. Blood was collected and processed to measure serum leptin using the mouse leptin ELISA kit (#90030, Crystal Chem, Inc.). Serum samples with hemolysis (one from each group) were excluded from leptin ELISA assay. The gonadal white adipose tissue, the inguinal white adipose tissue, and the interscapular brown adipose tissue were isolated and weighed.

Similarly, MpomcSRC-1-KO mice and their control littermates were weaned at week 4. These mice were singly housed and maintained on the standard chow (6.5% fat, #2920, Harlan-Teklad). All mice received tamoxifen injections (0.2 mg/g, i.p., twice) at 9 weeks of age. At the age of day 84, mice were switched to the HFD (60% fat, #D12492, Research Diets) for 30 days. Body weight and food intake were measured every 4 days. Body composition was determined using QMR on 30 days after HFD feeding.

*Food intake and energy expenditure*: To further characterize the food intake and energy expenditure of pomcSRC-1-KO mice, an independent male cohort (pomcSRC-1-KO mice and their control littermates) was weaned on the standard chow. At the age of 12 weeks, these mice were acclimated into the Comprehensive Laboratory Animal Monitoring System (CLAMS). Mice were housed individually at room temperature (22 °C) under an alternating 12:12-h light-dark cycle. After adaptation for 3 days, mice were subjected to a 2-day-chow–2-day-HFD protocol. Chow was replaced by HFD before the onset of dark cycle on day 3. Note that, the body weight and body composition were measured before the mice entered the CLAMS metabolic cages, and no difference was observed in body weight, fat mass, and lean mass.

Another male cohort (pomcSRC-1-KO mice and their control littermates) was weaned on the standard chow. At the age of 11 weeks, these mice were singly housed and at week 12, the chow diet was replaced by HFD. HFD intake was measured every 2 days for 10 days.

*Leptin-induced anorexia*: Male pomcSRC-1-KO mice and their control littermates (chow-fed) were briefly fasted for 2 h prior to the onset of dark cycle. These mice received intraperitoneal injections of saline or leptin (5 mg/kg in saline in a volume of 0.01 ml/g body weight) at 15 min prior to the dark cycle. The standard chow was provided at the onset of dark cycle. Food intake was measured 1, 4, and 24 h after food provision. Each mouse was tested with saline and leptin, administered in a counterbalanced order, with 4-day interval between the treatments.

*Electrophysiology*: For electrophysiological studies, *Pomc-CreER/Rosa26-tdTOMATO* (control) mice and *Pomc-CreER/Rosa26-tdTOMATO/SRC-1*[lox/lox] (MpomcSRC-1-KO) mice received tamoxifen inductions (0.2 mg/g, i.p., twice at 9 weeks of age) and fed on HFD for 1 week. *Pomc-CreER/Rosa26-tdTOMATO/ SRC-1*[L1376P/+] and their control littermates (*Pomc-CreER/Rosa26-tdTOMATO*) were also fed on HFD for 1 week followed by electrophysiology recording as described below. Briefly, at 9:00–9:30 am, these mice were deeply anesthetized with isofluorane and transcardially perfused with a modified ice-cold artificial cerebral spinal fluid (aCSF, in mM: 10 NaCl, 25 NaHCO₃, 195 Sucrose, 5 Glucose, 2.5 KCl, 1.25 NaH₂PO₄, 2 Na pyruvate, 0.5 CaCl₂, 7 MgCl₂)[47]. The mice were then decapitated, and the entire brain was removed. Brains was quickly sectioned in ice-cold aCSF solution (in mM: 126 NaCl, 2.5 KCl, 1.2 MgCl₂, 2.4 CaCl₂, 1 NaH₂PO₄, 11.1 Glucose, and 21.4 NaHCO₃)[23] saturated with 95% O₂ and 5% CO₂. Coronal sections containing the ARH (250 μm) was cut with a Microm HM 650V vibratome (Thermo Scientific). Then the slices were recovered in the aCSF[23] at 34 °C for 1 h.

Whole-cell patch clamp recordings were performed in the TOMATO-labeled mature Pomc neurons in the ARH visually identified by an upright microscope (Eclipse FN-1, Nikon) equipped with IR-DIC optics (Nikon 40× NIR). Signals were processed using Multiclamp 700B amplifier (Axon Instruments), sampled using Digidata 1440A and analyzed offline on a PC with pCLAMP 10.3 (Axon Instruments). The slices were bathed in oxygenated aCSF[23] (32–34 °C) at a flow rate of approximately 2 ml/min. Patch pipettes with resistances of 3–5 MΩ were filled with solution containing 126 mM K gluconate, 10 mM NaCl, 10 mM EGTA, 1 mM MgCl₂, 2 mM Na-ATP and 0.1 mM Mg-GTP (adjusted to pH 7.3 with KOH).

Current clamp was engaged to test neural firing frequency and RM at the baseline and after puff application of leptin (300 nM, 1 s). In some experiments, the aCSF solution also contained 1 μM TTX and a cocktail of fast synaptic inhibitors, namely bicuculline (50 μM; a GABA receptor antagonist), DAP-5 (30 μM; an NMDA receptor antagonist) and CNQX (30 μM; an AMPA receptor antagonist) to block the majority of presynaptic inputs. The values for RM and firing frequency were averaged within 2-min bin at the baseline or after leptin puff. The RM values were calculated by Clampfit 10.3 using the "analysis → statistic" function of the software. A neuron was considered depolarized or hyperpolarized if a change in membrane potential was at least 2 mV in amplitude and this response was observed after leptin application and stayed stable for at least 2 min. For the miniature inhibitory postsynaptic current (mIPSC) recordings, patch electrodes were filled with a recording solution that contained (in mM): 153.3 CsCl, 1.0 MgCl2, 5.0 EGTA, and 10.0 HEPES, pH of 7.20 with CsOH. CsCl was included to block potassium currents. Mg-ATP (3 mM) was added to the intracellular solution before recording. Glutamate receptor-mediated synaptic currents were blocked by 30 μM D-AP-5 and 30 μM CNQX in the external solution, along with 1 μM tetrodotoxin in the external solution blocking action potentials. Neurons were voltage-clamped at −70 mV during the recording.

At the end of recordings, lucifer yellow dye was included in the pipette solution to trace the recorded neurons and the brain slices were fixed with 4% formalin overnight and mounted onto slides. Cells were then visualized with the Leica DM5500 fluorescence microscope to identify post hoc the anatomical location of the recorded neurons in the ARH.

*Real-time PCR analyses*: Total RNA was isolated using TRIzol Reagent (Invitrogen) according to the manufacturer's protocol and reverse transcription reactions were performed from 2 μg of total RNA using a High-Capacity cDNA Reverse Transcription Kits (Invitrogen). cDNA samples were amplified on an CFX384 Real-Time System (Bio-Rad) using SsoADV SYBR Green Supermix (Bio-Rad). Correct melting temperatures for all products were verified after

amplification. Results were normalized against the expression of house-keeping gene-Cyclophilin. Primer sequences were listed in Supplementary Table 1.

*Immunoprecipitation (Co-IP) and immunoblotting*: The harvested hypothalami were lysed in lysis buffer (50 mM Tris–HCl, pH 8.0, 50 mM KCl, 20 mM NaF, 1 mM Na3VO4, 10 mM sodium pyrophosphate, 5 mM EDTA, and 0.5% Nonidet P-40) supplemented with protease inhibitors (1 mm phenylmethylsulfonyl fluoride, and 20 μg/ml each of leupeptin, aprotinin, and pepstatin). Lysates were cleared by centrifugation at 18,000 ×g for 10 min and used for immunoprecipitation or directly for immunoblotting. Equal amounts of tissue lysates were incubated with anti-Phospho-STAT3 (Tyr705) (D3A7) XP-Sepharose beads (Cell Signaling) or with a rabbit monoclonal SRC-1 (128E7) antibody (Cell Signaling) after preclearing for overnight and pulled down with Protein A/G agarose beads (Santa Cruz), respectively. Beads were washed three times with lysis buffer, and proteins were released from beads in SDS-sample buffer and analyzed by immunoblotting. For immunoblotting, protein samples were loaded onto SDS-polyacrylamide gels and transferred to a nitrocellulose membrane. The blot was probed with a rabbit monoclonal SRC-1 (128E7) antibody at 1:3000 (Cell Signaling), a rabbit monoclonal phospho-STAT3 (Tyr705) (D3A7) XP antibody at 1:2000 (Cell Signaling), or a monoclonal anti-β-Actin antibody (AC-15) at 1:10000 (Sigma). The secondary antibody was rabbit anti-mouse IgG or goat anti-rabbit IgG (Jackson ImmunoResearch), both at a 1:10,000 dilution, followed by development with the SuperSignal West Pico Chemiluminescent Substrate (Pierce).

*Chromatin immunoprecipitation assay (ChIP)*: Fresh isolated hypothalami were homogenized and cross-linked in 1% formaldehyde. Then, the cross-linked protein–DNA complexes were sonicated to a length between 200 and 500 bp. The total chromatin (1%) was saved as an "input" for later quantification. Complexes were pre-cleared and incubated with the Pierce Protein A/G Magnetic Beads (Thermo Scientific) and antibodies against STAT3 (sc-482; Santa Cruz) overnight at 4 °C. Subsequently, cross-linking was reversed by overnight incubation at 65 °C. DNAs were purified by phenol/chloroform extraction, ethanol precipitation and the enriched promoter fragments were measured by qPCR (primer sequences provided in Supplementary Table 1). Relative STAT3 promoter occupancy was adjusted to the background content of the negative control, and the initial chromatin input. The assays were repeated independently 3 times.

*Generation of SRC-1 constructs and expression plasmids*: The long form of SRC-1 containing a C-terminal Flag MYC tag was purchased from Origene (RC224812). The short form of SRC-1 was generated using the Q5 site-directed mutagenesis kit (NEB) using primers containing the sequence specific to the short form of SRC-1. The N-terminal HA tag was added using the Q5 site-directed mutagenesis kit (NEB) using primers containing the HA tag sequence. The short and long forms of SRC-1 was then cloned into the pCDNA3.1(+) vector using KpnI and XhoI restriction sites after PCR amplification of SRC-1 using primers flanking the Origene KpnI and XhoI sites. SRC-1 mutant constructs were generated using the Quickchange II XL site-directed mutagenesis kit (Agilent).

*In vitro protein interaction*: HEK293 (Human embryonic kidney 293) cells were transfected with either Flag-tagged transcriptional factor (hSTAT3 or hPPARγ), Flag-tagged human hormone receptor (ERα, VDR, THRβ or GR) or empty vector using lipofectamine 2000 (Invitrogen). Before harvest, cell were treated with leptin (at 200 ng/ml, 15 min, HARBOR-UCLA Research And Education Institute), or rosiglitazone (at 50 μM, ADIPOGEN), 17β-estradiol (at 0.2 μg/ml, Sigma, E2758), Vitamin D3 (Calcitriol at 0.2 μM, TOCRIS), dexamethasone (at 10 μM, Sigma, D4902) for 30 min. Cells were collected and lysed with cell lysis buffer: 50 mM Tris, 50 mM KCL, 10 mM EDTA, 1% NP-40, supplied with protease inhibitor cocktail (Roche) and phosphatase inhibitor cocktail A (Santa Cruz). The lysates were incubated with proper amount of anti-phospho-STAT3 sepharose beads (Cell Signaling, #4074) or anti-Flag-beads (Sigma) for 4 h at 4 °C. After wash, beads were aliquoted equally and incubated with comparable amounts of SRC-1 protein (wt or mutants) overnight, and the interacting protein was detected by Western-Blot. SRC-1 WT or mutants were expressed in HEK293 cells and the amount of the SRC-1 expressed was determined by Western-Blot before the protein interaction assay. Comparable amounts of SRC-1 (wt or mutants) in the same volume of cell lysates (compensated with the cell lysates from the cells transfected with empty vector) were used for the in vitro protein interaction. Except for the THRβ IP were equal amounts of total protein from SRC-1 WT and mutant lysates (determined by Bradford assay (Biorad)) were incubated with equal volumes of flag-tagged THRβ lysate overnight at 4 °C with 1 μM T3 thyroid hormone. THRβ was then immunoprecipitated using anti-Flag conjugated beads for 1 h which were washed 6 times with lysis buffer and eluted with LDS sample buffer before western blotting.

*Luciferase transcription activation assays*: To measure STAT3 activity on the POMC promoter, HEK293, Neuro 2A (mouse neuroblastoma cell line) and immortalized MEF cells (generated in J.X. lab) were cultured in Dulbecco's modified Eagle's medium supplemented with 10% fetal bovine serum (Atlanta), 100 IU/ml penicillin and 100 ng/ml streptomycin. Cells were seeded into a 24-well plate overnight and then transfected with 600 ng of the Pomc-luciferase reporter plasmid[66] or 300 ng Socs3-luciferase 6T1 reporter plasmid[12], combined with 100 ng of pRL-SV40 (Promega), 100 ng of pCR3.1-SRC-1 and/or 10 ng pRc/CMV-STAT3C plasmids or the control empty plasmids, according to the Lipofectamine LTX protocol (Invitrogen). Thirty hours post-transfection, the cells were lysed and the luciferase activity was measured using the Dual-Luciferase® Reporter Assay System (Promega) according to the manufacturer's instruction.

For leptin-induced Pomc-luciferase reporter assay, a fragment of the human POMC promoter (−949 to +416, relative to the transcription start site) was cloned into the pGL3 Luciferase Reporter Vector by using the primer pairs: 5′-TGTTCT AGTTGGGGGAACAGC-3′ and 5′-GCGCCCTTACCTGTCTCGG-3′. Neuro 2A cells were cultured in 48-well plate for overnight and then transfected with 0.1 μg human Pomc-luciferase reporter plasmid, 0.025 μg LepR and 0.05 μg hSRC-1 plasmid. Forty hours post-transfection, the cells were treated with 0.2 μg/ml Leptin for 20 min and then kept cultured in fresh media for 6 h. To test the effect of dominant negative STAT3 on leptin-induced POMC-luciferase reporter activity, the above protocol was modified by cotransfecting 10 ng of the dominant negative form of STAT3 (Y705F).

*Human studies*: The Genetics of Obesity Study (GOOS) is a cohort of 7000 individuals with severe early-onset obesity; age of obesity onset is less than 10 years[67,68]. Severe obesity is defined as a body mass index (weight in kilograms divided by the square of the height in meters) standard deviation score greater than 3 (standard deviation scores calculated according to the UK reference population). All studies were conducted in accordance with ethical regulations. The study protocol was reviewed and approved by the Cambridge Local Research Ethics Committee and each subject (or their parent for those under 16 years) provided written informed consent; minors provided oral consent.

Exome sequencing and targeted resequencing was performed in 2548 European ancestry individuals of the GOOS cohort (referred to as SCOOP) and in 1117 ancestry-matched controls[16]. Eleven rare variants (minor allele frequency <1%) in SRC-1 were identified in this study[16]; another 8 variants were identified in an earlier data release. Fifteen of these rare variants were identified in severely obese cases and 4 in the control dataset.

*Quantification and statistical analysis*: The minimal sample size was pre-determined by the nature of experiments. The actual sample size was indicated in each figure legend. The data are presented as mean ± SEM. Statistical analyses were performed using GraphPad Prism to evaluate normal distribution and variations within and among groups. Methods of statistical analyses were chosen based on the design of each experiment and are indicated in figure legends. $P < 0.05$ was considered to be statistically significant.

**Reporting summary**. Further information on experimental design is available in the Nature Research Reporting Summary linked to this article.

## Data availability

All relevant data are available from the authors. The source data underlying Figs. 1–4 and Supplementary Figs. 1–4 are provided as Source Data files. A Reporting Summary for this Article is available as a Supplementary Information file.

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

## Acknowledgements

Studies in cells and animals were supported by grants from the NIH (R01DK093587, R01DK101379, and R01DK117281 to Y.X.; R01DK114279 to Q.T., R01CA112403 and R01CA193455 to J.X.; R01HD07857, R01HD008818, and P01DK059820 to B.W.O'M.; P0101DK113954 to Y.X., J.X., and B.W.O'M.), USDA/CRIS (6250-51000-059-04S to Y.X.), American Heart Association (to Q.T. and Y.X.), Chinese National Natural Science Foundation (NSFC 81873558 to L.Z.). This work was partially supported by the Cancer Prevention and Research Institute of Texas (RP170005) and the NCI Cancer Center Support Grant (P30CA125123). Measurements of body composition, food intake, and energy expenditure in mice were performed in the Mouse Metabolic Research Unit (MMRU) at the USDA/ARS Children's Nutrition Research Center, Baylor College of Medicine, which is supported by funds from the USDA ARS (www.bcm.edu/cnrc/mmru). The Pomc-CreER mouse line was provided by Dr. Joel Elmquist (University of Texas Southwestern Medical Center); the *SRC-1^{lox/lox}* mouse line was provided by Dr. Pierre

Chambon (IGBMC/GIE-CERBM). The authors thank BCM Mouse Embryonic Stem Cell Core and Genetically Engineered Mouse Core for their assistance in generating the SRC-1[L1376P] knock-in mouse. The authors acknowledge the expert assistance of Mr. Firoz Vohra and the MMRU Core Director, Dr. Marta Fiorotto. Studies in humans were supported by the Wellcome Trust (A.A. v d K., T.M.C., I.B., I.S.F.) (098497/Z/12/Z; WT098051; 203513/Z/16/Z), NIHR Cambridge Biomedical Research Centre (I.S.F., I.B., S. O'R.), Bernard Wolfe Health Neuroscience Endowment (I.S.F.). T.M.C. was supported by a Philip Greenwood Clinical Research Fellowship and a Wellcome Trust Research Training Fellowship. The authors are indebted to the participants and their families for their participation and to the Physicians involved in the Genetics of Obesity Study (GOOS) (www.goos.org.uk); whole-exome sequencing was performed as part of the UK10K Consortium (a full list of investigators who contributed to the generation of the human genetics data as part of the UK10K consortium is available from www.UK10K.org.uk).

## Author contributions

Y.Y., A.A. v d K., Y.X., and I.S.F. conceived and designed the studies. Y.X. and I.S.F. supervised the project. Y.Y. and L.Z. performed and C.W., P.X., K.S., A.H.J., X.Y., L.L., and J.X. contributed to the experiments involving animals. Y.H. performed electrophysiological studies. Q.T. and B.W.O'M. participated in the design of the animal studies. A.A. v d K., T.M.C., J.M.K., and E.H. contributed to the human studies and with S.O'R. and I.S.F. to the recruitment of the GOOS cohort. T.M.C., L.K.J.S., and M.C.B. performed the molecular studies of human mutations. A.E.H., E.G.B., V.M., K.L.L., I.B., and I.S.F. contributed to the human genetic studies as part of work supported by the UK10K Consortium.

## Additional information

**Competing interests:** The authors declare no competing interests.

# UK10K Consortium
## Production group

Senduran Balasubramanian[4], Peter Clapham[4], Guy Coates[4], Tony Cox[4], Allan Daly[4], Petr Danecek[4], Yuanping Du[8], Richard Durbin[4], Sarah Edkins[4], Peter Ellis[4], Paul Flicek[4,9], Xiaosen Guo[8,10], Xueqin Guo[8], Liren Huang[8], David K. Jackson[4], Chris Joyce[4], Thomas Keane[4], Anja Kolb-Kokocinski[4], Cordelia Langford[4], Yingrui Li[8], Jieqin Liang[8], Hong Lin[8], Ryan Liu[11], John Maslen[4], Shane McCarthy[4], Dawn Muddyman[4], Michael A. Quail[4], Jim Stalker[4], Jianping Sun[12,13], Jing Tian[8], Guangbiao Wang[8], Jun Wang[8,10,14,15,16], Yu Wang[8], Kim Wong[4] & Pingbo Zhang[8]

## Cohorts group

Ines Barroso[2,4], Ewan Birney[9], Chris Boustred[17], Marie-Jo Brion[17], Lu Chen[4,18], Gail Clement[19], Petr Danecek[4], George Davey Smith[17], Ian N.M. Day[17], Aaron Day-Williams[4,20], Thomas Down[4,21], Ian Dunham[9], Richard Durbin[4], David M. Evans[17,22], Ghazaleh Fatemifar[17], Tom R. Gaunt[17], Matthias Geihs[4], Celia M.T. Greenwood[12,13,23,24], Deborah Hart[19], Audrey E. Hendricks[4,5], Bryan Howie[25], Jie Huang[4], Tim Hubbard[4,21], Pirro Hysi[19], Valentina Iotchkova[4,9], Yalda Jamshidi[26], John P. Kemp[17,22], Genevieve Lachance[19], Daniel Lawson[17], Monkol Lek[27], Margarida Lopes[4,28,29], Daniel G. MacArthur[27,30], Jonathan Marchini[28,31], Mangino Massimo[19,32], Iain Mathieson[33], Shane McCarthy[4], Yasin Memari[4], Sarah Metrustry[19], Josine L. Min[17], Alireza Moayyeri[19,34], Dawn Muddyman[4], Kate Northstone[17], Kalliope Panoutsopoulou[4], Lavinia Paternoster[17], John R.B. Perry[19,35], Lydia Quaye[19], J. Brent Richards[12,13,19,23,36], Susan Ring[17,37], Graham R.S. Ritchie[4,9], Stephan Schiffels[4], Hashem A. Shihab[17], So-Youn Shin[4,17], Kerrin S. Small[19], Mari´a Soler Artigas[38], Nicole Soranzo[4,18], Lorraine Southam[4,28], Timothy D. Spector[19], Beate St Pourcain[8,39,40], Gabriela Surdulescu[19], Ioanna Tachmazidou[4], Nicholas J. Timpson[17], Martin D. Tobin[38,41], Ana M. Valdes[19], Peter M. Visscher[22,42],

Louise V. Wain[38], Klaudia Walter[4], Kirsten Ward[19], Scott G. Wilson[19,43,44], Kim Wong[4], Jian Yang[22,42], Eleftheria Zeggini[4], Feng Zhang[19] & Hou-Feng Zheng[12,23,36]

## Neurodevelopmental disorders group

Richard Anney[45], Muhammad Ayub[46], Jeffrey C. Barrett[4], Douglas Blackwood[47], Patrick F. Bolton[48,49,50], Gerome Breen[48,49,50], David A. Collier[50,51], Nick Craddock[52], Lucy Crooks[4,53], Sarah Curran[48,54,55], David Curtis[56], Richard Durbin[4], Louise Gallagher[45], Daniel Geschwind[57], Hugh Gurling[56], Peter Holmans[52], Irene Lee[58], Jouko Lonnqvist[58], Shane McCarthy[4], Peter McGuffin[50], Andrew M. McIntosh[48], Andrew G. McKechanie[47,59], Andrew McQuillin[56,60], James Morris[4], Dawn Muddyman[4], Michael C. O'Donovan[52], Michael J. Owen[52], Aarno Palotie[4,61,62], Jeremy R. Parr[63], Tiina Paunio[64,65], Olli Pietilainen[4,64,61], Karola Rehnstrom[4], Sally I. Sharp[56], David Skuse[58], David St Clair[66], Jaana Suvisaari[64], James T.R. Walters[52] & Hywel J. Williams[52,67]

## Obesity group

Ines Barroso[2,4], Elena Bochukova[2], Rebecca Bounds[2], Anna Dominiczak[68], Richard Durbin[4], I. Sadaf Farooqi[2], Audrey E. Hendricks[4,5], Julia Keogh[2], Gaelle Marenne[4], Shane McCarthy[4], Andrew Morris[69], Dawn Muddyman[4], Stephen O'Rahilly[2], David J. Porteous[70], Blair H. Smith[71], Ioanna Tachmazidou[4], Eleanor Wheeler[4] & Eleftheria Zeggini[4]

## Rare disease group

Saeed Al Turki[4,72], Carl A. Anderson[4], Dinu Antony[73], Ines Barroso[2,4], Phil Beales[73], Jamie Bentham[74], Shoumo Bhattacharya[74], Mattia Calissano[75], Keren Carss[4], Krishna Chatterjee[2], Sebahattin Cirak[75,76], Catherine Cosgrove[74], Richard Durbin[4], David R. Fitzpatrick[77], James Floyd[4,78], A. Reghan Foley[75], Christopher S. Franklin[4], Marta Futema[79], Detelina Grozeva[80], Steve E. Humphries[79], Matthew E. Hurles[4], Shane McCarthy[4], Hannah M. Mitchison[73], Dawn Muddyman[4], Francesco Muntoni[75], Stephen O'Rahilly[2], Alexandros Onoufriadis[21], Victoria Parker[2], Felicity Payne[4], Vincent Plagnol[81], F. Lucy Raymond[80], Nicola Roberts[80], David B. Savage[2], Peter Scambler[73], Miriam Schmidts[73,82], Nadia Schoenmakers[2], Robert K. Semple[2], Eva Serra[4], Olivera Spasic-Boskovic[80], Elizabeth Stevens[75], Margriet van Kogelenberg[4], Parthiban Vijayarangakannan[4], Klaudia Walter[4], Kathleen A. Williamson[77], Crispian Wilson[80] & Tamieka Whyte[75]

## Statistics group

Antonio Ciampi[13], Celia M.T. Greenwood[12,13,23,24], Audrey E. Hendricks[4,5], Rui Li[12,23,36], Sarah Metrustry[19], Karim Oualkacha[83], Ioanna Tachmazidou[4], ChangJiang Xu[12,13] & Eleftheria Zeggini[4]

## Ethics group

Martin Bobrow[82], Patrick F. Bolton[48,49,50], Richard Durbin[4], David R. Fitzpatrick[77], Heather Griffin[84], Matthew E. Hurles[4], Jane Kaye[84], Karen Kennedy[4,85], Alastair Kent[86], Dawn Muddyman[4], Francesco Muntoni[75], F. Lucy Raymond[78], Robert K. Semple[2], Carol Smee[4], Timothy D. Spector[19] & Nicholas J. Timpson[17]

## Incidental findings group

Ruth Charlton[87], Rosemary Ekong[88], Marta Futema[79], Steve E. Humphries[79], Farrah Khawaja[89], Luis R. Lopes[90,91], Nicola Migone[92], Stewart J. Payne[93], Vincent Plagnol[81], Rebecca C. Pollitt[94], Sue Povey[88], Cheryl K. Ridout[95], Rachel L. Robinson[87], Richard H. Scott[73], Adam Shaw[96], Petros Syrris[90], Rohan Taylor[89] & Anthony M. Vandersteen[97]

## Management committee

Jeffrey C. Barrett[4], Ines Barroso[2,4], George Davey Smith[17], Richard Durbin[4], I. Sadaf Farooqi[2], David R. Fitzpatrick[77], Matthew E. Hurles[4], Jane Kaye[85], Karen Kennedy[4], Cordelia Langford[4], Shane McCarthy[4], Dawn Muddyman[4], Michael J. Owen[52], Aarno Palotie[4,61,62], J. Brent Richards[12,13,19,23], Nicole Soranzo[4,18], Timothy D. Spector[19], Jim Stalker[4], Nicholas J. Timpson[17] & Eleftheria Zeggini[4]

## Lipid meta-analysis group

Antoinette Amuzu[98], Juan Pablo Casas[90,98], John C. Chambers[34], Massimiliano Cocca[99,100], George Dedoussis[101], Giovanni Gambaro[102], Paolo Gasparini[99,100,103], Tom R. Gaunt[17], Jie Huang[4], Valentina Iotchkova[4,9], Aaron Isaacs[104], Jon Johnson[105], Marcus E. Kleber[106], Jaspal S. Kooner[107], Claudia Langenberg[108], Jian'an Luan[108], Giovanni Malerba[109], Winfried Marz[110,111,112], Angela Matchan[4], Josine L. Min[17], Richard Morris[113], Børge G. Nordestgaard[114,115], Marianne Benn[114,115], Susan Ring[37], Robert A. Scott[109], Nicole Soranzo[4,18], Lorraine Southam[4,28], Nicholas J. Timpson[17], Daniela Toniolo[116], Michela Traglia[116], Anne Tybjaerg-Hansen[115,117], Cornelia M. van Duijn[104], Elisabeth M. van Leeuwen[104], Anette Varbo[114,115], Peter Whincup[118], Gianluigi Zaza[119], Eleftheria Zeggini[4] & Weihua Zhang[100]

[8]BGI-Shenzhen, Shenzhen 518083, China. [9]European Molecular Biology Laboratory, European Bioinformatics Institute, Wellcome Trust Genome Campus, Hinxton, Cambridge CB10 1SD, UK. [10]Department of Biology, University of Copenhagen, Ole Maaløes Vej 5, DK-2200 Copenhagen, Denmark. [11]BGI-Europe, London EC2M 4YE, UK. [12]Lady Davis Institute, Jewish General Hospital, Montreal, Quebec H3T 1E2, Canada. [13]Department of Epidemiology, Biostatistics and Occupational Health, McGill University, Montreal, Quebec H3A 1A2, Canada. [14]Princess Al Jawhara Albrahim Center of Excellence in the Research of Hereditary Disorders, King Abdulaziz University, P.O. Box 80200Jeddah 21589, Saudi Arabia. [15]Macau University of Science and Technology, Avenida Wai long, Taipa, Macau 999078, China. [16]Department of Medicine and State Key Laboratory of Pharmaceutical Biotechnology, University of Hong Kong, 21 Sassoon Road, Hong Kong, Hong Kong. [17]MRC Integrative Epidemiology Unit, School of Social and Community Medicine, University of Bristol, Oakfield House, Oakfield Grove, Clifton, Bristol BS8 2BN, UK. [18]Department of Haematology, University of Cambridge, Long Road, Cambridge CB2 0PT, UK. [19]The Department of Twin Research & Genetic Epidemiology, King's College London, St Thomas' Campus, Lambeth Palace Road, London SE1 7EH, UK. [20]Computational Biology & Genomics, Biogen Idec, 14 Cambridge Center, Cambridge, MA 02142, USA. [21]Department of Medical and Molecular Genetics, Division of Genetics and Molecular Medicine, King's College London School of Medicine, Guy's Hospital, London SE1 9RT, UK. [22]University of Queensland Diamantina Institute, Translational Research Institute, Brisbane, Queensland 4102, Australia. [23]Department of Human Genetics, McGill University, Montreal, Quebec H3A 1B1, Canada. [24]Department of Oncology, McGill University, Montreal, Quebec H2W 1S6, Canada. [25]Adaptive Biotechnologies Corporation, Seattle, WA 98102, USA. [26]Human Genetics Research Centre, St George's University of London, London SW17 0RE, UK. [27]Analytic and Translational Genetics Unit, Massachusetts General Hospital, Boston, MA 02114, USA. [28]Wellcome Trust Centre for Human Genetics, Roosevelt Drive, Oxford OX3 7BN, UK. [29]Illumina Cambridge Ltd, Chesterford Research Park, Cambridge CB10 1XL, UK. [30]Program in Medical and Population Genetics, Broad Institute of Harvard and MIT, Cambridge, MA 02142, USA. [31]Department of Statistics, University of Oxford, 1 South Parks Road, Oxford OX1 3TG, UK. [32]National Institute for Health Research (NIHR) Biomedical Research Centre at Guy's and St Thomas' Foundation Trust, London SE1 9RT, UK. [33]Department of Genetics, Harvard Medical School, Boston, MA 02115, USA. [34]The Department of Epidemiology and Biostatistics, Imperial College London, St Mary's campus, Norfolk Place, Paddington, London W2 1PG, UK. [35]MRC Epidemiology Unit, University of Cambridge School of Clinical Medicine, Box 285, Institute of Metabolic Science, Cambridge Biomedical Campus, Cambridge CB2 0QQ, UK. [36]Department ofMedicine, Jewish General Hospital, McGill University, Montreal, Quebec H3A 1B1, Canada. [37]ALSPAC & School of Social and Community Medicine, University of Bristol, Oakfield House, Oakfield Grove, Clifton, Bristol BS8 2BN, UK. [38]Departments of Health Sciences and Genetics, University of Leicester, Leicester LE1 7RH, UK. [39]School of Oral and Dental Sciences, University of Bristol, Lower Maudlin Street, Bristol BS1 2LY, UK. [40]School of Experimental Psychology, University of Bristol, 12a Priory Road, Bristol BS8 1TU, UK. [41]National Institute for Health Research (NIHR) Leicester Respiratory Biomedical Research Unit, Glenfield Hospital, Leicester LE3 9QP, UK. [42]Queensland Brain Institute, University of Queensland, Brisbane, Queensland 4072, Australia. [43]School of Medicine and Pharmacology, University of Western Australia, Perth, Western Australia 6009, Australia. [44]Department of Endocrinology and Diabetes, Sir Charles Gairdner Hospital, Nedlands, WA 6009, Australia. [45]Department of Psychiatry, Trinity Centre for Health Sciences, St James Hospital, James's Street, Dublin 8, Ireland. [46]Division of Developmental Disabilities, Department of Psychiatry, Queen's

University, Kingston, ON N6C 0A7, Canada. [47]ivision of Psychiatry, The University of Edinburgh, Royal Edinburgh Hospital, Edinburgh EH10 5HF, UK. [48]Department of Child Psychiatry, Institute of Psychiatry, Psychology and Neuroscience, King's College London, 16 De Crespigny Park, London SE5 8AF, UK. [49]NIHR BRC for Mental Health, Institute of Psychiatry, Psychology and Neuroscience and SLaM NHS Trust, King's College London, 16 De Crespigny Park, London SE5 8AF, UK. [50]Social, Genetic and Developmental Psychiatry Centre, Institute of Psychiatry, Psychology and Neuroscience, King's College London, Denmark Hill, London SE5 8AF, UK. [51]Lilly Research Laboratories, Eli Lilly & Co. Ltd., Erl Wood Manor, Sunninghill Road, Windlesham GU20 6PH, UK. [52]MRC Centre for Neuropsychiatric Genetics & Genomics, Institute of Psychological Medicine & Clinical Neurosciences, School of Medicine, Cardiff University, Cardiff CF24 4HQ, UK. [53]Sheffield Diagnostic Genetics Service, Sheffield Childrens' NHS Foundation Trust, Western Bank, Sheffield S10 2TH, UK. [54]University of Sussex, Brighton BN1 9RH, UK. [55]Sussex Partnership NHS Foundation Trust, Swandean, Arundel Road, Worthing BN13 3EP, UK. [56]University College London (UCL), Molecular Psychiatry Laboratory, Division of Psychiatry, Gower Street, London WC1E 6BT, UK. [57]UCLA David Geffen School of Medicine, Los Angeles, CA 90095, USA. [58]Behavioural and Brain Sciences Unit, UCL Institute of Child Health, London WC1N 1EH, UK. [59]The Patrick Wild Centre, The University of Edinburgh, Edinburgh EH10 5HF, UK. [60]University College London (UCL), UCL Genetics Institute, Darwin Building, Gower Street, London WC1E 6BT, UK. [61]Institute for Molecular Medicine Finland (FIMM), University of Helsinki, Helsinki FI-00014, Finland. [62]Program in Medical and Population Genetics and Genetic Analysis Platform, The Broad Institute of MIT and Harvard, Cambridge, MA 02132, USA. [63]Institute of Neuroscience, Henry Wellcome Building for Neuroecology, Newcastle University, Framlington Place, Newcastle upon Tyne NE2 4HH, UK. [64]National Institute for Health and Welfare (THL), Helsinki FI-00271, Finland. [65]University of Helsinki, Department of Psychiatry, Helsinki FI-00014, Finland. [66]Institute of Medical Sciences, University of Aberdeen, Aberdeen AB25 2ZD, UK. [67]The Centre for Translational Omics – GOSgene, UCL Institute of Child Health, London WC1N 1EH, UK. [68]Institute of Cardiovascular and Medical Sciences, University of Glasgow, Wolfson Medical School Building, University Avenue, Glasgow G12 8QQ, UK. [69]Usher Institute of Population Health Sciences and Informatics, University of Edinburgh, 9 Little FranceRoad, Edinburgh EH16 4UX, UK. [70]Centre for Genomic and Experimental Medicine, Institute of Genetics and Experimental Medicine, University of Edinburgh, Western General Hospital, Crewe Road, Edinburgh EH4 2XU, UK. [71]Mackenzie Building, Kirsty Semple Way, Ninewells Hospital and Medical School, Dundee DD2 4RB, UK. [72]Department of Pathology, King Abdulaziz Medical City, P.O. Box 22490Riyadh 11426, Saudi Arabia. [73]Genetics and Genomic Medicine and Birth Defects Research Centre, UCL Institute of Child Health, London WC1N 1EH, UK. [74]Department of Cardiovascular Medicine and Wellcome Trust Centre for Human Genetics, Roosevelt Drive, Oxford OX3 7BN, UK. [75]Dubowitz Neuromuscular Centre, UCL Institute of Child Health & Great Ormond Street Hospital, London WC1N 1EH, UK. [76]Institut fur Humangenetik, Uniklinik Köln, Kerpener Strasse 34, 50931 Köln, Germany. [77]MRC Human Genetics Unit, MRC Institute of Genetics and Molecular Medicine, at the University of Edinburgh, Western General Hospital, Edinburgh EH4 2XU, UK. [78]The Genome Centre, John Vane Science Centre, Queen Mary, University of London, Charterhouse Square, London EC1M 6BQ, UK. [79]Cardiovascular Genetics, BHF Laboratories, Rayne Building, Institute of Cardiovascular Sciences, University College London, London WC1E 6JJ, UK. [80]Academic Laboratory of Medical Genetics, Box 238, Lv 6 Addenbrooke's Treatment Centre, Addenbrooke's Hospital, Cambridge CB2 0QQ, UK. [81]University College London (UCL) Genetics Institute (UGI) Gower Street, London WC1E 6BT, UK. [82]Genetics Department, Radboudumc and Radboud Institute for Molecular Life Sciences (RIMLS), Geert Grooteplein 25, Nijmegen 6525 HP, The Netherlands. [83]Department of Mathematics, Université de Québec À Montréal, Montréal, QC H3C 3P8, Canada. [84]HeLEX – Centre for Health, Law and Emerging Technologies, Nuffield Department of Population Health, University of Oxford, Old Road Campus, Oxford OX3 7LF, UK. [85]National Cancer Research Institute, Angel Building, 407 St John Street, London EC1V 4AD, UK. [86]Genetic Alliance UK, 4D Leroy House, 436 Essex Road, London N1 3QP, UK. [87]Leeds Genetics Laboratory, St James University Hospital, Beckett Street, Leeds LS9 7TF, UK. [88]University College London (UCL) Department of Genetics, Evolution & Environment (GEE), Gower Street, London WC1E 6BT, UK. [89]SW Thames Regional Genetics Lab, St George's University, Cranmer Terrace, London SW17 0RE, UK. [90]Institute of Cardiovascular Science, University College London, Gower Street, London WC1E 6BT, UK. [91]Cardiovascular Centre of the University of Lisbon, Faculty of Medicine, University of Lisbon, Avenida Professor Egas Moniz, 1649-028 Lisbon, Portugal. [92]Department of Medical Sciences, University of Torino, 10124 Torino, Italy. [93]North West Thames Regional Genetics Service, Kennedy-Galton Centre, Northwick Park Hospital, Watford Road, Harrow HA1 3UJ, UK. [94]Connective Tissue Disorders Service, Sheffield Diagnostic Genetics Service, Sheffield Children's NHS Foundation Trust, Western Bank, Sheffield S10 2TH, UK. [95]Molecular Genetics, Viapath at Guy's Hospital, London SE1 9RT, UK. [96]Clinical Genetics, Guy's & St Thomas' NHS Foundation Trust, London SE1 9RT, UK. [97]Maritime Medical Genetics Service, 5850/5980 University AvenuePO Box 9700Halifax, NS B3K 6R8, Canada. [98]London School of Hygiene and Tropical Medicine, Keppel Street, London WC1E 7HT, UK. [99]Medical Genetics, Institute for Maternal and Child Health IRCCS "Burlo Garofolo", 34100 Trieste, Italy. [100]Department of Medical, Surgical and Health Sciences, University of Trieste, 34100 Trieste, Italy. [101]Department of Nutrition and Dietetics, School of Health Science and Education, Harokopio University, Athens 17671, Greece. [102]Division of Nephrology and Dialysis, Institute of Internal Medicine, Renal Program, Columbus-Gemelli University Hospital, Catholic University, 00168 Rome, Italy. [103]Experimental Genetics Division, Sidra, P.O. Box 26999Doha, Qatar. [104]Genetic Epidemiology Unit, Department of Epidemiology, Erasmus MC, Rotterdam 3000 CA, Netherlands. [105]Department of Quantitative Social Science, UCL Institute of Education, University College London, 20 Bedford Way, London WC1H 0AL, UK. [106]Vth Department of Medicine, Medical Faculty, Mannheim 68167, Germany. [107]National Heart and Lung Institute, Imperial College London, London W12 0NN, UK. [108]MRC Epidemiology Unit, University of Cambridge School of Clinical Medicine, Institute of Metabolic Science, Cambridge Biomedical Campus, Cambridge CB2 0QQ, UK. [109]Biology and Genetics, Department of Life and Reproduction Sciences, University of Verona, 37134 Verona, Italy. [110]Clinical Institute of Medical and Chemical Laboratory Diagnostics, Medical University of Graz, Graz 8036, Austria. [111]Synlab Academy, Synlab Services GmbH, D-68161 Mannheim, Germany. [112]Medical Clinic V (Nephrology, Hypertensiology, Rheumatology, Endocrinolgy, Diabetology), Mannheim Medical Faculty, Heidelberg University, Mannheim 68167, Germany. [113]School of Social and Community Medicine, Canynge Hall, 39 Whatley Road, Bristol BS8 2PS, UK. [114]Department of Clinical Biochemistry and The Copenhagen General Population Study, Herlev and Gentofte Hospital, Copenhagen University Hospital, Herlev 2730, Denmark. [115]The Faculty of Health and Medical Sciences, University of Copenhagen, Copenhagen 2200, Denmark. [116]Division of Genetics and Cell Biology, San Raffaele Scientific Institute, Milan 20132, Italy. [117]Department of Clinical Biochemistry KB3011, Rigshospitalet, Copenhagen University Hospital, Blegdamsvej 9, DK-2100 Copenhagen, Denmark. [118]Population Health Research Institute, St George's University of London, London SW17 0RE, UK. [119]Renal Unit, Department of Medicine, University of Verona, 37126 Verona, Italy

