## [Peer Review File · Nature Communications]

Reviewers' comments:

Reviewer #3 (Remarks to the Author):

THanks for the responses to the previous reviews. As before i focused on the human genetics. I note that the authors have reduced the emphasis on the human genetics, and note the more nuanced reporting of the 738 allele (even though i think this looks more like WT than LOF across all assays).

The response re the 1272 allele wasnt entirely clear, but i assume they did indeed see the 1272 allele (as they should have givne the frequency) but excluded it from further analysis as it was considered not to meet some arbitrary definition of "rare". That may well be an entirely reasonable decision, though without functional analysis of the variant one cant be sure. I would like to see some specific comment about the justification for not including the 1272 allele in the manuscript, since that decision is likely to have had a substantial impact on the apparent strength of the human genetic findings (the count differential would not have looked anywhere near as impressive had 1272 alleles been counted.)

Reviewer #4 (Remarks to the Author):

I appreciate the way the authors addressed my comments on electrophysiology and AgRP neurons. While debating the the potential shortcomings of in vivo recordings of neurons by measuring calcium signals is of intellectual value, it really goes against common sense. First, convincing data is available to show that this calcium signals reflect the speed by which various hormones act on feeding (e.g., gherkin versus leptin). Second, at present, it is difficult to argue for the superiority of a technique that relies on decapitation and further processing of tissue before recording is done. I would definitely drop those experiments If they are crucial for the paper's main point, I remain less enthusiastic. I accept the points they raised about AgRP neurons, which nevertheless deserve a discussion.

Reviewer #5 (Remarks to the Author):

The arguments made by Farooqi, et al., are correct- there are many potential reasons that could explain the rather modest and slow effects of leptin on Ca⁺² transients in POMC cells in vivo (i.e., the imperfect correlation between Ca⁺² and electrical activity and the limitations inherent in fiber photometry for mixed populations of neurons, as is the case here). The electrophysiologic control of a subset of POMC neurons by leptin has been repeatedly demonstrated by many groups. Their electrophysiologic findings provide useful support to their overall conclusions.

It can't hurt to put a sentence in about AgRP neurons, but the authors are also correct that leptin action on POMC neurons plays a known role in the control of energy balance, and the magnitude of this role correlates well with the magnitude of their effects.

Re: Steroid Receptor Coactivator-1 Modulates the Function of Pomc Neurons and Energy Homeostasis

Reviewers' comments:

Reviewer #3 (Remarks to the Author):

THanks for the responses to the previous reviews. As before i focused on the human genetics. I note that the authors have reduced the emphasis on the human genetics, and note the more nuanced reporting of the 738 allele (even though i think this looks more like WT than LOF across all assays).

The response re the 1272 allele wasnt entirely clear, but i assume they did indeed see the 1272 allele (as they should have givne the frequency) but excluded it from further analysis as it was considered not to meet some arbitrary definition of "rare". That may well be an entirely reasonable decision, though without functional analysis of the variant one cant be sure. I would like to see some specific comment about the justification for not including the 1272 allele in the manuscript, since that decision is likely to have had a substantial impact on the apparent strength of the human genetic findings (the count differential would not have looked anywhere near as impressive had 1272 alleles been counted.)

We thank the reviewer for their comments. SRC-1 P1272S was not reported in our manuscript because, given its minor allele frequency (MAF; 2.52% in gnomAD non-Finnish European Controls), it falls above our threshold for calling rare variants which was <1% as detailed in the revised methods. On review, the MAF of this variant is 3.16% in cases and 3.45% in controls. It is interesting that this variant is more common in our data than in gnomAD. However, this could be for a variety of reasons, such as differences in variant calling, analysis and/or collapsed ancestries in the web version. Of interest, whilst one might assume that variants of this frequency are unlikely to have functional consequences, this variant has been studied in cells previously (Hartmeier et al, Mol Endocrinology 2012;26(2):220-7), where SRC-1 P1272S was shown to disrupt a putative glycogen synthase 3 (GSK3) β phosphorylation site and showed a decreased ability to co-activate Estrogen Receptor compared to wild-type SRC-1 in multiple cell lines. Additionally, we note that there are some low frequency and many rare variants in SRC-1 in publically available databases. In light of the reviewer's comments, we have added the following additional comments on page 11".

"We identified 15 rare heterozygous variants in SRC-1 in 16 severely obese individuals and 4 rare variants in controls. Notably, there are several low frequency and many rare variants in this gene in publically available databases (<http://gnomad.broadinstitute.org/>). Some of these low frequency variants have been shown to have functional consequences, for example, P1272S (MAF: 3.16% in cases, 3.45% in controls; 2.52% in gnomAD non-Finnish European Controls) disrupts a putative

glycogen synthase 3 (GSK3) β phosphorylation site and has been shown to exhibit reduced ability to co-activate Estrogen Receptor in multiple cell lines (Hartmeier et al, Mol Endocrinology 2012;26(2):220-7). Genetic studies in larger numbers of cases and controls with functional studies of all variants identified will be needed to establish whether variants that result in a loss of function when tested in cells are more likely to be found in severely obese individuals than in controls. In this study, the variants found in obese individuals, but not those found in controls, were associated with impaired interaction with pSTAT3 and reduced POMC reporter activity in cells, predominantly through a dominant negative effect. Given the challenges associated with studying such rare variants, and to directly test whether rare human SRC-1 variants contribute to Pomc neuron function and/or energy homeostasis, we generated a knock-in mouse model of a human variant which results in a severe loss of function in cells, SRC-1^{L1376P}.”

Reviewer #4 (Remarks to the Author):

I appreciate the way the authors addressed my comments on electrophysiology and AgRP neurons. While debating the the potential shortcomings of in vivo recordings of neurons by measuring calcium signals is of intellectual value, it really goes against common sense. First, convincing data is available to show that this calcium signals reflect the speed by which various hormones act on feeding (e.g., gherkin versus leptin). Second, at present, it is difficult to argue for the superiority of a technique that relies on decapitation and further processing of tissue before recording is done. I would definitely drop those experiments If they are crucial for the paper's main point, I remain less enthusiastic. I accept the points they raised about AgRP neurons, which nevertheless deserve a discussion.

Reviewer #5 (Remarks to the Author):

The arguments made by Farooqi, et al., are correct- there are many potential reasons that could explain the rather modest and slow effects of leptin on Ca²⁺ transients in POMC cells in vivo (i.e., the imperfect correlation between Ca²⁺ and electrical activity and the limitations inherent in fiber photometry for mixed populations of neurons, as is the case here). The electrophysiologic control of a subset of POMC neurons by leptin has been repeatedly demonstrated by many groups. Their electrophysiologic findings provide useful support to their overall conclusions.

It can't hurt to put a sentence in about AgRP neurons, but the authors are also correct that leptin action on POMC neurons plays a known role in the control of energy balance, and the magnitude of this role correlates well with the magnitude of their effects.

We thank the reviewer for their consideration. We have now added the following clarification regarding the contribution of both Pomc and Agrp neurons to feeding on page 11.

“Additionally, leptin-responsive Agrp neurons have been shown to play a major role in energy homeostasis (Xu et al., Nature. 2018 Apr;556(7702):505-509).”

REVIEWERS' COMMENTS:

Reviewer #3 (Remarks to the Author):

I agree that this final set of revisions captures all outstanding questions that I have.

REVIEWERS' COMMENTS:

Reviewer #3 (Remarks to the Author):

I agree that this final set of revisions captures all outstanding questions that I have.

Response: There is no further issues raised by referees.